# Farmers’ Adaptive Behaviors to Heavy Metal-Polluted Cultivated Land in Mining Areas: The Influence of Farmers’ Characteristics and the Mediating Role of Perceptions

**DOI:** 10.3390/ijerph19116718

**Published:** 2022-05-31

**Authors:** Yong Chen, Yaqi Liang, Hao Zhou, Qiaozhi Wang, Yanzhong Liu

**Affiliations:** 1College of Resource and Environmental Engineering, Wuhan University of Science and Technology, No.947 Heping Avenue, Wuhan 430080, China; chenyong68@wust.edu.cn (Y.C.); lyq1791933149@163.com (Y.L.); wangqiaozhi@wust.edu.cn (Q.W.); liuyanzhong@wust.edu.cn (Y.L.); 2National Key Laboratory of Environmental Protection Mining and Metallurgy Resource Utilization and Pollution Control, Wuhan 430080, China; 3Hubei Provincial Key Laboratory of Efficient Utilization and Agglomeration of Metallurgical Mineral Resources, Wuhan 430080, China

**Keywords:** farmers’ adaptive behaviors, heavy metal pollution, mediating effect, factor analysis, hierarchical regression

## Abstract

Heavy metal pollution in cultivated land poses a serious threat to environmental health and farmers’ livelihoods. As the direct user of cultivated land, understanding farmers’ adaptive behavior to heavy metal pollution, and its influencing factors, can provide insight and information relevant for decision-making, so as to better manage the hazards and risks of heavy metal pollution. We proposed a conceptual framework of “farmers’ characteristics-perceptions-adaptive behaviors”. Factor analysis and mediation effect analysis were used to explore the influence of characteristics and perceptions on adaptive behaviors. The data of 278 farmers in a typical mining area in Daye, China, show that local farmers perceive the hazards of heavy metal pollution, but their adaptive behaviors are hindered to a certain extent. The results of the mediation effect analysis show that perceptions of health impact, self-efficacy, and adaptive cost play a partial mediating role in the impact of characteristics on adaptive behaviors. In addition, the influence of the “factor of dependence on farmland” and the “factor of obstacles to action” on adaptive behavior have no significant relationship with perception levels. By comparing the influencing factors, we found that although farmers’ perceptions have mediating effects between characteristics and adaptive behaviors, characteristics still play a decisive role in adaptive behaviors.

## 1. Introduction

Heavy metal pollution (HMP) of cultivated land is becoming a severe ecological and social problem in China [1,2]. According to the National Soil Pollution Survey Bulletin, the over-standard rate of heavy metals in cultivated land in China is as high as 19.4% [3]. Due to HMP, more than 10 million tons of grain production are reduced, and 12 million tons of grains contaminated, each year [4]. HMP has the characteristics of accumulation, persistence, and irreversibility [5,6]. When heavy metals accumulate in the soil to a certain level, it not only reduces the quality of cultivated land and affects crop growth, but also threatens human health and well-being through the food chain [7,8]. Thus, widespread public concern about food safety and health risks are aroused [9].

Mineral resources are an important material basis for social and economic development [10]. As one of the world’s largest metal producers and consumers, China has diversified and large-scale mineral resources. The proven reserves of mineral resources account for 12% of the world’s total mineral resources [11]. Chinese scholars identified that the main sources of soil HMP in China were sewage irrigation, sludge application, and mining and smelting of minerals [1]. Among them, the mining industry was considered to be one of the most important sources of HMP. Heavy metals produced in the process of mining, beneficiation, and smelting often cause large-scale soil pollution in the region, through the atmosphere, surface runoff, and groundwater [12].

Strict measures were taken by the Chinese government to alleviate HMP in soil. During the “Twelfth Five-Year Plan” period (2011–2015), more than RMB 21 billion was invested in the treatment of HMP. In 2016, the “Action Plan for Soil Pollution Prevention and Control” was issued. It proposed that the safe utilization rate of polluted cultivated land will reach more than 95% by 2030, and the pilot of cultivated land fallow compensation will be carried out in some areas with serious pollution [13]. However, the implementation of the work was restricted, due to the large area of polluted cultivated land, the high cost, and the long period of remediation. By 2020, only about 6.67 thousand km^2^ of cultivated land polluted by heavy metals were improved. The actual effect of cultivated land fallow was not satisfactory either, and the scale of fallow land was far from reaching the government’s goal [14]. Part of the heavy metal-contaminated farmland is still used.

The existing research on soil HMP focuses on the evaluation of pollution status and remediation technology. However, there is a lack of research on farmers. In China, farmers are the main micro-organizational agents of agricultural production. Since the implementation of the “household contract responsibility system”, the right to use cultivated land was decentralized to farmers [15]. Farmers became the direct users of cultivated land, and the direct bearers of risk control. Understanding their adaptive behaviors toward heavy metal-polluted cultivated land, and the influencing factors, can provide insights and information relevant for decision-making to better manage the hazards and risks of HMP in cultivated land. It is also very crucial in reducing the vulnerability of farmers, securing their livelihood, and formulating future strategies to control heavy metal-polluted cultivated land. But there are few scientific studies on farmers’ living or production behaviors on heavy metal-contaminated land.

In this paper, 278 farmers in the cultivated land polluted by heavy metals in typical mining areas of Daye, China were considered as the research population. We studied the adaptive behaviors of farmers under the threat of HMP and its influencing factors, to make two contributions to the current research. First, our study provides a new case for the study of adaptation to global environmental change, and gives a support basis for the Chinese government to formulate the management policy of heavy metal-contaminated farmland at the farmer level. Second, we proposed a conceptual framework of “farmers’ characteristics-perceptions-adaptive behaviors”, and examined the influence of characteristics and perceptions on adaptive behaviors.

## 2. Materials and Methods

### 2.1. Conceptual Framework

Adaptation refers to the process by which people make adjustments according to actual or expected environmental changes, and their impacts to reduce or avoid harm, or exploit beneficial opportunities [16]. With the intensification of conflicts between humans and the environment, the study of adaptive behaviors has become a frontier area and a core component of environmental change research [17,18,19]. Akcaoz and Ozkan indicate that agriculture is a uniquely high-risk and uncertain industry. Farmers must make decisions in an environment where risks are constantly changing [20]. Understanding the key factors affecting farmers’ adaptation process can help governments formulate adaptation policies to ensure sustainable agricultural development [21,22]. In recent years, more attention was paid to the adaptive behaviors of farmers in response to climate change, and primarily focused on climate change perception, adaptive capacity, and adaptive measures [23,24,25]. Moreover, because adaptation is a complex and multidimensional process [26], the current understanding of farmers’ adaptation to environmental changes, and the factors that affect them, is still very limited [27,28].

Perception is considered to be the basis of adaptation [29,30]. Deressa et al. divide the adaptation process of farmers to environmental change into two stages: perceiving environmental change and adopting adaptation measures [31]. Lorenzoni et al. propose a three-stage theory, which postulates that the behavioral–psychological process of individual involvement in environmental matters consists of three stages: perceiving environmental change, perceiving the effects of change, and adapting through appropriate behaviors [32]. Bohensky et al. argue that there are three stages for individuals to adapt to environmental changes: observation, perception, and action. The latter stage must be based on the previous stage [33]. Different levels of perception lead to differences in adaptation strategies, so farmers with more obvious perceptions of environmental changes and risks are more likely to take action [34]. However, some scholars found that higher levels of risk perception does not always translate into positive adaptive behaviors [35]. Farmers’ assessments of self-efficacy, adaptive costs, and other factors are mediating factors that impede the transformation of risk perception into adaptive behavior [36]. Grothmann and Patt developed the model of private proactive adaptation to climate change (MPPACC) and define two perceived psychological process variables in the model: risk appraisal and adaptation appraisal. They believe that the two variables are affected by factors such as farmer’s characteristics and social background [37]. Age, race, education, living area, income level, agricultural experience, and other characteristics may affect farmers’ environmental risk perceptions [38,39,40].

Above all, there is a complex nonlinear relationship between farmers’ characteristics, perceptions, and adaptive behaviors. Nevertheless, existing studies mainly discuss the impact of farmers’ perceptions on adaptive behaviors, or take the unity of farmers’ characteristics and perceptions as independent variables to explore the influence of characteristics and perceptions on adaptive behaviors [27,41,42,43,44]. Few studies focus on the mediating role of farmers’ perception between their characteristics and adaptive behavior. Therefore, we constructed a conceptual framework of “farmers’ characteristics-perceptions-adaptive behaviors”, as shown in Figure 1. Then, we hypothesized that farmers’ characteristics have a direct effect on adaptive behaviors, while perceptions mediate the relationship between them. A further explanation of the framework is shown below. 

### 2.2. Study Area and Survey Process

#### 2.2.1. Study Area

Three villages seriously affected by HMP in Daye, China were selected as the study areas (Figure 2). Daye is a famous mining city in China, with more than 3000 years of mining and smelting history. In 2012, it was listed as a national key demonstration zone for HMP remediation by the Ministry of Environmental Protection. As early as the 1980s, the problem of HMP in Daye City emerged, such as deterioration of water quality, crop yield reduction (or even no harvest), variation in the quality of agricultural products, and an increase in tumor patients. Over the past 30 years, rapid economic development led to a continuous increase in the demand for mineral resources. High-intensity and extensive mining and smelting of mineral resources aggravated the HMP of soil, water, and agricultural products around the mining area [45]. The agricultural production and physical health of farmers are seriously affected [46].

#### 2.2.2. Survey Process

In China, farmers’ cultivated land is often located around their residences. We selected 11 villages in Daye City that were relatively seriously affected by HPM. Participatory rural appraisal (PRA) was used to investigate farmers. Two pre-surveys were conducted in October and November 2020, to identify common behaviors adapted to HMP of cultivated land. To ensure that the questionnaire was in line with the actual situation and easily understood by the farmers, we supplemented and improved the questionnaire in accordance with the pre-surveys. The formal survey was conducted in January 2021. According to the stratified sampling method, the household survey was carried out with households as units. To make the results reliable, we provided assistance to each respondent during the survey, and provided them with a return of 30 RMB after answering the questions. A total of 300 questionnaires were collected and the final sample size was 278, with an effective rate of 92.67%. 

### 2.3. Farmers’ Adaptive Behaviors

By visiting farmers and consulting experts familiar with the local environment, this research identified six categories of relatively common adaptive behaviors for HMP on cultivated land, as shown in Table 1. From the perspective of cultivated land utilization, we called the four types of adaptive behaviors (i.e., plowing or reshaping the soil, adjusting irrigation patterns, adjusting crop varieties, and changing land-use patterns) pollution mitigation behaviors (Behav-M). These behaviors mean that farmers are still trying to alleviate and utilize the cultivated land polluted by heavy metals. Abandoning farming and trying to move away were named pollution avoidance behaviors (Behav-A), because adopting these behaviors means that farmers are not going to mitigate and use cultivated land polluted by heavy metals.

### 2.4. Selection of Farmers’ Characteristics and Perceptions

#### 2.4.1. Farmers’ Characteristics

Farmers’ characteristics were primarily considered from the perspective of the individual and household characteristics. Unlike in the traditional Chinese household registration, the householder in this paper refers to the person who determines a household’s agricultural production and livelihood preferences, to a certain extent [47]. Individual characteristics primarily cover the age and education level of the householder, which can reflect their non-agricultural employment ability and information acquisition ability [48]. Household characteristics reflect the socioeconomic base and agricultural production conditions. Referring to relevant research [44,49,50], we selected 9 indicators, including the age and education level of the householder, family size, population dependency ratio, agricultural income share, nonfarm labor force share, arable land per capita (m^2^), distance from the pollution source (Km), and pollution time (year), as shown in Table 2. Among them, the distance from the pollution source and the pollution time are assigned a value of 1, 2, or 3, according to the distance between the farmer’s group and the mine pollution source and the mining time, respectively.

#### 2.4.2. Farmers’ Perceptions

According to the MPPACC model [37], we investigated farmers’ risk perception and adaptation perception of HMP, as shown in Appendix A. The risk perceptions in the MPPACC model include possibility perception and severity perception. Since the study areas have been polluted by heavy metals for a long time, farmers hold a very positive and consistent view on the possibility of HMP. In the survey, we found that farmers mainly paid attention to the impact of HMP on agricultural production and health. Therefore, the risk perception of farming (Risk-F) and health (Risk-H) are considered as the risk perceptions in this study. Adaptive perceptions include adaptive efficacy perception (Adapt-A), self-efficacy perception (Adapt-S), and adaptive cost perception (Adapt-C) [37]. It is difficult for farmers to accurately express their adaptive cost perceptions due to the lack of professional knowledge and skills in mitigating heavy metal pollution on cultivated land The development of urbanization in China since the reform and opening up (reform and opening up is a policy of domestic reform and opening up that China began to implement at the Third Plenary Session of the Eleventh Central Committee in December 1978) provided farmers with a great number of higher-income and non-farm employment opportunities, so farmers no longer considered the direct costs of production when simply making agricultural decisions. Our study drew on the concept of opportunity cost in economics to investigate adaptive cost perception [51]. We quantified farmers’ adaptive cost perception in terms of their willingness to spend time and money to mitigate HMP. The stronger the willingness, the lower the adaptive cost perception. For farmers’ perceptions, we set five questions, as shown in Appendix A. For the closed-ended questions, they were quantified according to a five-point Likert scale.

### 2.5. Mediation Effect Analysis Method

According to the mediating effect model developed by Baron and Kenny [52], we used hierarchical regression to explore the mediating effects of farmers’ perceptions between characteristics and adaptive behaviors. The first stage (Equation (1)): testing the effect of characteristics on adaptive behaviors. If $\delta_{1}$ is significant, we proceed to the second stage; (Equation (2)): exploring the influence of characteristics on perceptions. If $\beta_{1}$ is significant, go to the third stage; (Equation (3)): estimating whether characteristics and perceptions are related to adaptive behaviors. If both $\gamma_{1}$ and $\gamma_{2}$ are significant, there is a partial mediating effect, and if $\gamma_{2}$ is significant but $\gamma_{1}$ is not, there is a full mediating effect. Otherwise, there is no mediating effect.
(1)$$AdaptB=\delta_{0}+\delta_{1}FarC+\varepsilon_{1}$$
(2)$$FarP=\beta_{0}+\beta_{1}FarC+\varepsilon_{2}$$
(3)$$AdaptB=\gamma_{0}+\gamma_{1}FarC+\gamma_{2}FarP+\varepsilon_{3}$$
where *AdaptB* denotes farmers’ adaptive behaviors, *FarC* denotes farmers’ characteristics, and *FarP* denotes farmers’ perceptions. Since *AdaptB* is a binary categorical data, logistic regression is more applicable to Equations (1) and (3) than least squares (OLS), so Equations (1) and (3) are rewritten as Equations (4) and (5), respectively [53].
(4)$$AdaptB=LogitP(AdaptB=1|FarC)=\ln\frac{P(AdaptB=1|FarC)}{P(AdaptB=0|FarC)}=\delta_{0}+\delta_{1}FarC+\varepsilon_{1}$$
(5)$$AdaptB=LogitP(AdaptB=1|FarP,FarC)=\ln\frac{P(AdaptB=1|FarP,FarC)}{P(AdaptB=0|FarP,FarC)}=\gamma_{0}+\gamma_{1}FarC+\gamma_{2}FarP+\varepsilon_{3}$$

Generally, the value of mediating effect can be expressed as $\beta_{1}*\gamma_{2}$. However, Equations (2) and (3) in this paper were estimated by ordinary least squares (OLS) and logistic regression, respectively, leading to $\beta_{1}$, and $\gamma_{2}$ on different scales. The regression coefficients of the original equation need to be standardized to quantify the mediating effects [54]. Therefore, we used Mplus to construct the mediating effects model to obtain the standardized coefficients.

## 3. Results

### 3.1. Descriptive Statistics

Among the 278 valid questionnaires, we conducted statistical analysis on the basic information of the respondents, as shown in Table 2. Among them, 56.83% are male and 74.28% are over 45 years old. Only 17.27% of farmers received a high school education or above. The majority of farmers have a household size of three or above, and the labor force in the household is under heavy pressure to raise children and the elderly. Over half of them have less than 20% of agricultural income. Nearly 30% of families have a non-farm labor force of more than 60%. Approximately 33% of the respondents have less than 200 m^2^ arable land, which is lower than the average level of Daye City. In addition, about 41% of farmers’ cultivated land is less than 1 km away from the pollution source, and more than half of the cultivated land has been polluted for more than 20 years. The basic information of the respondents is in line with the characteristics of rural China. It reflects the situation of farmers in the study area.

#### 3.1.1. Adaptive Behaviors

As shown in Figure 3, statistics show that 60.8% of farmers take at least one adaptive behavior, 44.2% take pollution mitigation behaviors (Behav-M), and 27.3% adopt pollution avoidance behaviors (Behav-A). As Figure 4 shows, the adoption rate of each adaptive behavior is not high. Among the mitigation behaviors, the adoption rates of plowing or reshaping the soil (Behav-M1), adjusting irrigation mode (Behav-M2), and adjusting crop variety (Behav-M3) are less than 20%, and the adoption rate of changing land-use patterns (Behav-M4) is only 8.99%. On the contrary, the adoption rate of trying to move away (Behav-A1) is the highest, reaching 23.38%, indicating that farmers have a strong willingness to escape in the face of HMP in cultivated land.

#### 3.1.2. Farmers’ Characteristics

In this study, nine variables of farmers’ characteristics were selected. We used factor analysis (FA) to extract common factors. Based on SPSS 24.0, data were tested by KMO and Bartlett tests [55,56]. Then, principal component analysis and maximum variance rotation were used to complete the factor analysis, as shown in Table 3.

Factor 1 focuses on the pollution time and the distance from the pollution source, so it is named the “factor of pressure from mines”. The proportion of agricultural income and arable land per capita have higher loads on factor 2, which reflects the importance of agricultural production in households, and it is called the “factor of dependence on farmland”. Factor 3 primarily reflects the learning ability and adaptability of farmers, so it is designated as the “factor of householder’s quality”. Factor 4 refers to the family size and nonfarm labor share. The higher the score, the greater the pressure in life, which is entitled the “factor of obstacles to action”.

#### 3.1.3. Farmers’ Perceptions 

From Figure 4, it can be seen that farmers’ risk perceptions are relatively strong. Both the perceptions of farming impact (RiskP-F) and health impact (RiskP-H) of HMP in cultivated land are at a high level, and the latter is more prominent. The overall levels of adaptive perceptions are moderate or low. Among them, the level of adaptive cost perception (AdaptP-C) is the highest, and there are large fluctuations among individuals (standard deviation = 1.73). The level of self-efficacy perception (AdaptP-S) is the lowest and relatively stable (standard deviation = 0.97), indicating that farmers have insufficient confidence in self-adapting to HMP in cultivated land.

### 3.2. The Mediating Effect of Perception

#### 3.2.1. Hierarchical Regression 

According to the method in Section 2.4, we tested the mediating effect of farmers’ perceptions, as shown in Table 4. First, we tested the effect of farmers’ characteristics on adaptive behaviors (including mitigation and avoidance behaviors). The results are shown in models 1 and 2. Second, in models 3–7 we examine the influence of farmers’ characteristics on perceptions. Finally, the impacts of farmers’ perceptions and characteristics on adaptive behaviors are shown in models 8 and 9.

For pollution mitigation behaviors (Behav-M), in the first stage, we find that the “factor of pressure from mines” (Factor 1) and the “factor of dependence on farmland” (Factor 2) have significant effects on Behav-M. In the second stage, from model 6, we observe a significant and positive effect of Factor 1 on self-efficacy perception (AdaptP-S). According to the third stage, in model 8, there are significant positive effects of Factor 1 and AdaptP-S on Behav-M. In addition, risk perception of farming (RiskP-F) has a positive effect on Behav-M. It is revealed that Factor 2 and RiskP-F have direct positive effects on Behav-M, and AdaptP-S plays a partially mediating role in the relationship between Factor 1 and Behav-M.

For pollution avoidance behaviors (Behav-A), in the first stage, we identify significant effects of the “factor of pressure from mines” (Factor 1), the “factor householder’s quality” (Factor 3), and the “factor of obstacles to action” (Factor 4) on Behav-A. In the second stage, model 6 shows that there is a significant positive effect of Factor 1 on the perception of health impact (RiskP-H). From model 7, it is known that there is a significant and positive effect of Factor 3 on the adaptive cost perception (AdaptP-C). According to the third stage, in model 9, Factor 1 and RiskP-H have significant positive effects on Behav-A. In addition, risk perception of health (RiskP-H) has a positive effect on Behav-A. It is shown that Factor 4 has a significant negative effect on Behav-A and RiskP-H has a significant positive effect on Behav-A; RiskP-H partially mediates the relationship between Factor 1 and Behav-A; and AdaptP-C plays a partially mediating role in the relationship between Factor 3 and Behav-A.

Mplus was used to calculate the effect values of the three mediating paths, as shown in Table 5. The mediating effect of AdaptP-S is 0.031, accounting for 20.38% of the total effect. That means that 20.38% of the effect of Factor 1 on Behav-M is via AdaptP-S. The mediating effects of RiskP-H and AdaptP-C are 0.019 and 0.023, accounting for 11.04% and 11.42% of the total effect, respectively.

In addition, we took the Bootstrap test to further examine the reliability of the mediating effects [57,58]. Table 4 indicates that the three mediating effects are significant at the 5% and 10% levels, respectively, revealing that there are indeed mediating effects for these three farmer perceptions.

#### 3.2.2. Robustness Check 

To test the robustness of the results, the logistic regression model was converted into the OLS regression model. As shown in Table 6, the two regression models passed the same variables and the signs of the coefficients are identical, indicating that the results are robust.

## 4. Discussion

### 4.1. Strong Risk Perception and Low Self-Efficacy Perception

In the study areas, farmers have strong risk perceptions on HMP, including RiskP-F and RiskP-H. In the survey, we found that only 19.4% of farmers obtained information about HMP through the government. Their knowledge about HMP mainly came from farming experience, neighborhood discussion, and the media. These make it easy to form a relatively consistent view of HMP, and its risks within a village. The phenomenon that crops did not germinate, grow, or even die was common in the polluted cultivated land, which made farmers firmly believe that HMP has a great impact on farming. In comparison, RiskP-H is stronger. On the one hand, mining activities were carried out there for more than 20 years. In recent years, the incidence of cancer increased, and the incidence of the population started to be younger. The reports about “cancer villages” make them link cancer with HMP, which makes them very worried. On the other hand, with the development of China’s rural economy, the material living standards of farmers greatly improved. The majority of people’s thoughts changed from “solving adequate food and clothing” to “pursuing a better and healthy life”. Farmers are particularly concerned about health problems.

The AdaptP-S is generally low. In areas rich in mineral resources, mining is often the pillar industry of local development, which not only brings huge financial revenue to local governments, but also creates a large number of employment opportunities (Antoci et al., 2019). Local governments seek a balance between environmental protection and economic development (Ranängen and Lindman, 2018). To avoid conflicts between farmers and mining enterprises, local governments rarely disclose HMP to the public. There is a greater lack of corresponding knowledge and skills training. Low AdaptP-S discourages farmers from proactively adopting adaptive behaviors [59,60]. The lack of knowledge and skills makes local farmers negative and helpless when facing HMP. They want to seek help. So, 16.2% of farmers have the experience of seeking compensation from mining enterprises, and 30.6% seek compensation from local government.

### 4.2. Passive and Barely Ineffective Adaptive Behavior

Although 44.2% of farmers took Behav-M, and were still trying to mitigate and utilize the cultivated land polluted by heavy metals., due to the invisibility of HMP [2], it cannot be observed such as water eutrophication and other pollution. The lack of knowledge and skills makes farmers’ behaviors often passive, empirical, and constrained by natural environmental conditions. It is usually spread by word of mouth, based on the experience of residents in the village, and is not widely disseminated. The regional natures of behaviors result in the low adoption rate of each behavior.

It is found in the survey that the behavior of plowing or replacing the soil generally occurs on the cultivated land eroded by tailings or mine sludge. The black tailings or sludge makes farmers intuitively feel that the growth of crops would be affected. Changing the irrigation mode generally occurs in the cultivated land near polluted rivers. Long-term experience taught them that using the polluted river for irrigation makes the crops unable to grow normally, or even die. To this end, farmers tried to find new sources of water, such as diverting water from farther places, or building ponds. Limited by the natural environment, not all farmers could find suitable water sources. In this case, some of them could only try to plant different crops, adjust the utilization of land to plant trees, or build fish ponds to raise fish. However, due to the HMP caused by mining and metallurgical fume, farmers had no choice but to report to the government. In the study areas, due to the massive expropriation of farmers’ land for the development of mineral resources, the per capita cultivated land area of many farmers is less than 300 m^2^. For the sake of livelihood, farmers with high-dependence on cultivated land have to try to use the land with serious HMP, although they know that the output of the cultivated land is very low, and may even fail to harvest.

According to our investigation, the effect of mitigation behaviors is not ideal. Due to the accumulation and irreversibility of HMP [6], the cultivated land that was plowed or had an adjusted irrigation mode still had a low yield and high production cost. Even if the land was used for planting trees, the growth of seedlings was slow, and the mortality was high. We tested and evaluated a random sample of crops in the study areas. The results show that the comprehensive carcinogenic risk coefficients of heavy metals ingested by adults and children through staple food (rice) and vegetables are higher than the safety value. It means farmers’ daily consumption of self-produced agricultural products will pose the carcinogenic risk of heavy metals. 

### 4.3. The Mediation role of Perception and the Determinant Role of Characteristics

The results of the hierarchical regression indicate that both farmers’ characteristics and perceptions have important influences on their adaptive behaviors to HMP on cultivated land. 

From the perspective of farmers’ characteristics, the “factor of pressure from mines” (Factor 1) has a significant positive effect on both Behav-M and Behav-A. It is related to the socioeconomic background of the “mining-agricultural combination area”. The stress of mining on agricultural development has gradually become prominent, and farmers have to try to adjust production decisions to adapt to the changes in the agricultural environment [61]. While farmers in good condition will choose to abandon farming or even move away from polluted areas, similar to Factor 1, the “factor of dependence on farmland” (Factor 2) also has a significant positive effect on Behav-M. For farmers with a high-dependence on cultivated land, agriculture is often the economic pillar of their families. These families lack means of livelihood other than agricultural production. When they face HMP, they have to take measures to maintain agricultural production, and the basic livelihood of households. As can be seen from model 8 (Table 4), the coefficient of Factor 2 has the largest effect on Behav-M, indicating that the dependence on cultivated land is the most important factor promoting the adoption of Behav-M by farmers. The “factor of householder’s quality” (Factor 3) has a significant positive effect on Behav-A. Younger and better-educated people have more non-farm employment opportunities. In the context of rapid urbanization, some Chinese farmers prefer to work, or even live, in the cities [62]. Moreover, this trend is exacerbated in mining areas [63]. Confronted with the HMP of cultivated land, householders with high-quality learning ability and adaptability are more likely to abandon agricultural production, or move away from their current residence, because they have the economic basis to support these behaviors [64]. The “factor of obstacles to action” (Factor 4) has a significant negative effect on Behav-A, which is opposite to Factor 3. Households with larger action damping have more elders and children. It negatively affects the average income of family members, and reduces the marginal income of the labor force [65]. From model 9 (Table 4), we can see that the coefficient of Factor 4 has the largest effect on Behav-A, indicating that it is the most important factor that hinders Behav-A.

In terms of farmers’ perceptions, farming impact perception (RiskP-F), health impact perception (RiskP-H), self-efficacy perception (AdaptP-S), and adaptive cost perception (AdaptP-C) all have a positive and significant impact on farmers’ adaptive behaviors. RiskP-F promotes Behav-M; when farmers perceive a threat to their production, they try to take measures to mitigate the current situation. RiskP-H plays a mediating role between factor 1 and Behav-A; farmers gradually perceive the health hazards of HMP with the aggravation of pollution [66], which makes them have the idea of staying away from pollution. AdaptP-S plays a mediating role between factor 1 and Behav-M; long-term experience with heavy metal-polluted cultivated land enables farmers to conclude certain countermeasures, such as adjusting crop varieties and finding new sources of irrigation. These measures can increase their confidence in the face of HMP. Even though the effects of these measures are not clear to farmers, it does not prevent them from improving their perception of self-efficacy, then promoting the adoption of Behav-M. AdaptP-C plays a mediating role between factor 3 and Behav-A; householders with high-quality learning ability and adaptability can earn a much higher income than agricultural production in cities, and they have a higher opportunity cost to adopt mitigation behaviors. So, they are more inclined to adopt Behav-A.

Surprisingly, the effects of tillage dependence factors and action resistance factors on adaptive behavior have no significant relationship with perception level. For the former, as Manalo et al. conclude: those who are firmly committed to their identity as rice farmers will find ways to continue growing rice anyway, while those who are not firmly committed to their identity as rice farmers may leave the rice fields if they find it impossible to grow them due to drought [67]. For the latter, currently, the cost of education for children in China is increasing year by year, and the rural retirement system is still inadequate [68]. The economic pressure of supporting the elderly and raising children makes them incapable of abandoning farming. Meanwhile, there are no more funds to support relocation. Except for economic pressure, there is another important factor that hinders their avoidance behaviors, which is called “old farmer’s homeland attachment” in this paper. In rural China, some old farmers are reluctant to leave their long-established places and abandon their farmland. They prefer to maintain their current living conditions rather than take the “uncertain” risk of heavy metal pollution. 

The factors affecting farmers’ adaptive behaviors are complex. This study focuses on the effects of farmers’ characteristics on adaptive behaviors, and the impact of farmers’ perceptions on this relationship. Farmers’ characteristics can reflect household resources and production endowments. They play a decisive role in farmers’ adaptation to HMP in cultivated land. Farmers’ perceptions reflect farmers’ psychological factors. The results show that the perceptions of health impact, self-efficacy, and adaptation cost play a partial mediating role in the influence of farmers’ characteristics on adaptive behaviors.

## 5. Conclusions

To explain the mechanism of farmers’ adaptive behaviors towards heavy metal-polluted cultivated land in mining areas, we proposed a conceptual framework of “farmers’ characteristics-perceptions-adaptive behaviors”. Based on 278 farmers’ data in the mining areas of Daye, China, a hierarchical regression approach was used to analyze the influence of farmers’ characteristics and perceptions on adaptive behaviors, and the mediating role of farmers’ perceptions between farmers’ characteristics and adaptive behaviors.

Results show that farmers can perceive the harm of HMP in cultivated land, and have strong perceptions of farming impact and health impact, but farmers’ self-efficacy perception is very low, because they have very limited access to HMP knowledge and skills. A total of 60.8% of farmers adopt at least one adaptive behavior; 44.2% of farmers adopt pollution mitigation behavior; and 27.3% of them adopt pollution avoidance behavior, and the adoption rate of each adaptive behavior is not high. From the results of hierarchical regression, self-efficacy perception mediates the relationship between the “factor of pressure from mines” and pollution mitigation behaviors; health impact perception mediates the relationship between the “factor of pressure from mines” and pollution avoidance behaviors; and adaptive cost perception mediates the relationship between the “factor of householder’s quality” and pollution avoidance behaviors. The “factor of dependence on farmland” and the “factor of obstacles to action” are the most important influences on pollution mitigation behaviors and pollution avoidance behaviors, respectively. The former has a positive effect and the latter a negative effect. Farmers’ perceptions can affect their adaptive behaviors, and play a mediating role in the impact of farmers’ characteristics on behavior, but farmers’ characteristics play a decisive role. For those farmers who are highly dependent on cultivated land, taking pollution mitigation behavior is only for their need to maintain family livelihood, but has nothing to do with perception.

This study believes that there is an urgent need for comprehensive planning and coordination of the treatment and utilization of heavy metal-polluted cultivated land. First, strengthening the monitoring of HMP, and organizing professional skill training and production guidance, in order to improve farmers’ understanding and adaptability to HMP. Second, to help farmers better adapt, the government should give financial support to effectively mitigate HMP. For seriously polluted villages, it is necessary to organize the conversion of farmland and compensate farmers as soon as possible, or the mining enterprises should compensate in the form of land acquisition. For heavily polluted villages, the overall relocation of villages is the best choice. Third, when formulating relevant measures, differentiated countermeasures should be taken for different groups of farmers. Poor farmers with high-dependence on cultivated land may need the most attention.

This study still has some limitations, in that we only focus on farmers in a typical mining area in central China. In the follow-up study, we will conduct a comparative study in other regions of mining areas, to verify the accuracy and reliability of our findings.

## Figures and Tables

**Figure 1 ijerph-19-06718-f001:**
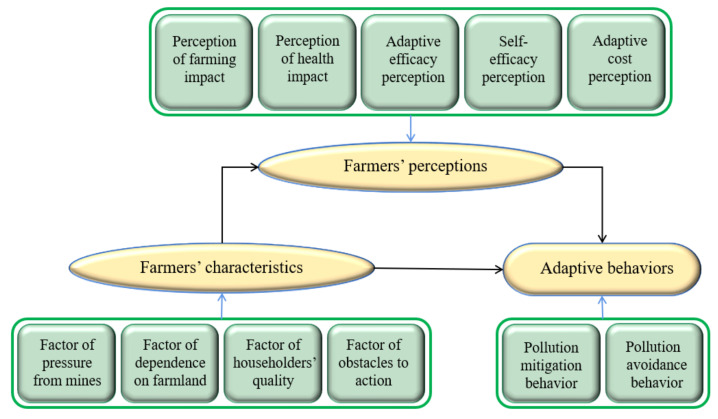
Conceptual framework of “farmers’ characteristics-perceptions-adaptive behaviors”.

**Figure 2 ijerph-19-06718-f002:**
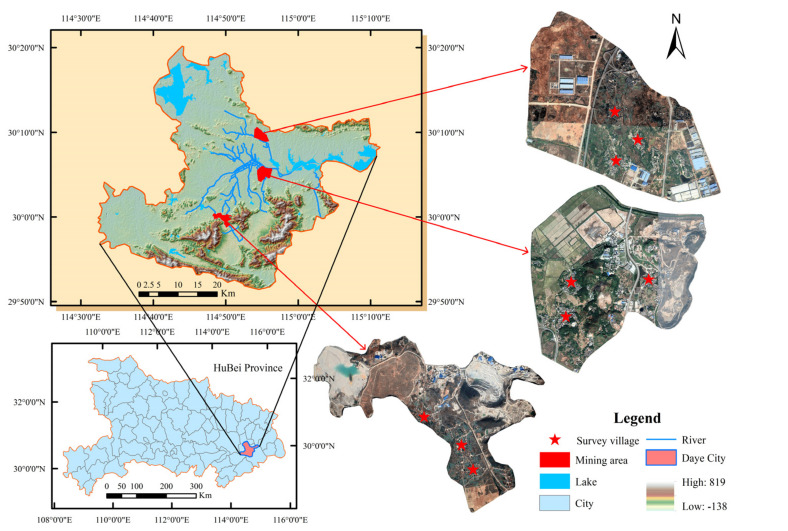
The location of the study areas.

**Figure 3 ijerph-19-06718-f003:**
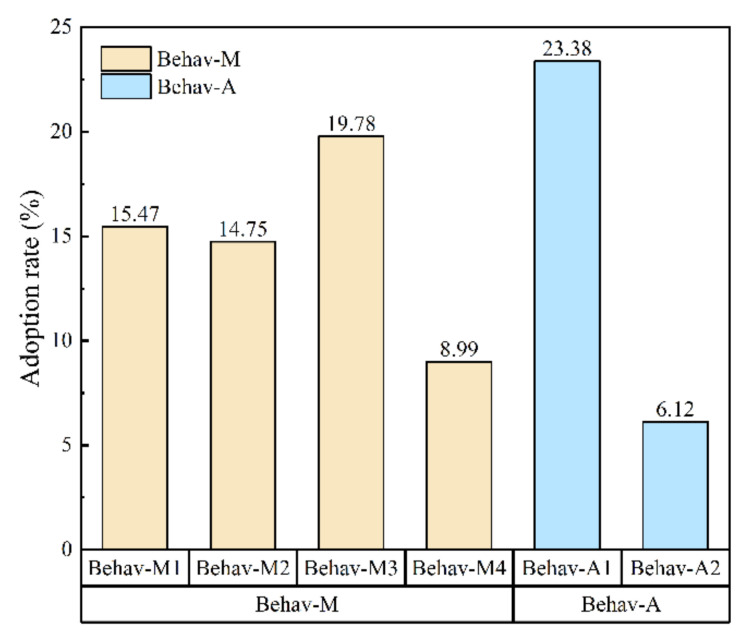
Farmers’ adaptive behaviors to heavy metal pollution in cultivated land.

**Figure 4 ijerph-19-06718-f004:**
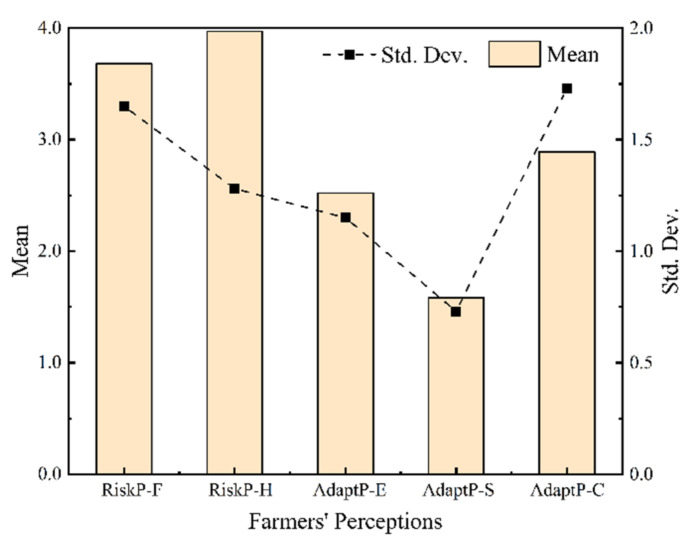
Levels of perceptions on HMP in cultivated land.

**Table 1 ijerph-19-06718-t001:** Farmers’ adaptive behaviors toward heavy metal-polluted cultivated land.

Adaptive Behaviors		Description
Pollution mitigation behaviors (Behav-M)	Plow or replace the soil (Behav-M1)	Plow or change the soil eroded by sewage or tailings
	Adjust irrigation patterns (Behav-M2)	Irrigate or find new water sources by observing the water regime
	Adjust crop varieties (Behav-M3)	Select heavy metal pollution-tolerant varieties according to experience or media publicity
	Change land-use patterns (Behav-M4)	Planting trees or fish ponds instead of crops
Pollution avoidance behaviors (Behav-A)	Try to move away (Behav-A1)	Abandoning crop cultivation because of the perception of severe pollution
	Abandon farming (Behav-A2)	Trying to move out because of the feeling of serious pollution

**Table 2 ijerph-19-06718-t002:** Descriptive statistics of respondents’ basic information.

Farmers’ Characteristics	Class	QTY	PCT(%)	S.D.
Gender	Female	120	43.17	0.50
	Male	158	56.83	
Age	≤45	70	25.18	10.74
	45~60	134	48.20	
	>60	74	26.62	
Education level	Primary school	111	39.93	0.76
	Junior high school	119	42.81	
	High school and above	48	17.26	
Family size	≤3	55	19.78	1.97
	3~5	126	45.33	
	>5	97	34.89	
Population dependency ratio	≤0.5	84	30.22	0.53
	0.5~1	120	43.17	
	>1	74	26.62	
Agricultural income share	≤20%	163	58.63	0.95
	20~40%	60	21.58	
	>40%	55	19.78	
Nonfarm labor force share	≤20%	79	28.42	0.32
	20%~60%	115	41.37	
	>60%	84	30.22	
Arable land per capita	≤200	93	33.45	0.43
	200~466.67	101	36.33	
	>466.67	84	30.22	
Distance from pollution source	≤1	114	41.01	0.19
	1~2	115	41.37	
	>2	49	17.63	
Pollution time	≤10	88	31.65	0.89
	10~20	43	15.47	
	>20	147	52.88	

**Table 3 ijerph-19-06718-t003:** Results of factor analysis of farmers’ characteristics.

Farmer Characteristics	Characteristic Factors				Data Inspection
	Factor 1	Factor 2	Factor 3	Factor 4	
Age	0.159	−0.104	0.806	−0.155	KMO = 0.61 Sig. = 0.000
Education level	0.029	0.115	0.861	0.063	
Family size	−0.150	−0.086	−0.204	0.782	
Population dependency ratio	0.039	−0.034	−0.051	0.865	
Agricultural income	0.096	0.909	−0.014	−0.012	
Nonfarm labor share	−0.149	−0.217	0.150	0.563	
Arable land per capita(mu)	0.045	0.828	0.043	−0.297	
Distance from the pollution source	0.890	0.050	0.069	−0.069	
Pollution time	0.877	0.094	0.119	−0.127	

**Table 4 ijerph-19-06718-t004:** The mediating effect of perceptions between characteristics and behaviors.

Variables	First Stage		Second Stage					Third Stage	
	Model 1:	Model 2:	Model 3:	Model 4:	Model 5:	Model 6:	Model 7:	Model 8:	Model 9:
	Behav-M	Behav-A	RiskP-F	RiskP-H	AdaptP-E	AdaptP-S	AdaptP-C	Behav-M	Behav-A
Factor 1	0.287 ** (0.129)	0.379 ** (0.151)	0.087 (0.099)	0.112 *** (0.076)	−0.064 (0.068)	0.174 *** (0.043)	0.029 (0.103)	0.237 * (0.134)	0.278 * (0.156)
Factor 2	0.427 *** (0.130)	−0.079 (0.141)	−0.050 (0.099)	−0.123 *** (0.076)	−0.14 ** (0.068)	−0.016 (0.043)	−0.039 (0.103)	0.474 *** (0.137)	−0.052 (0.147)
Factor 3	0.173 (0.125)	0.437 *** (0.140)	0.094 (0.099)	0.076 (0.076)	0.177 *** (0.068)	0.204 *** (0.043)	0.169 *** (0.103)	0.083 (0.132)	0.362 ** (0.149)
Factor 4	−0.138 (0.125)	−0.370 *** (0.144)	0.018 (0.099)	−0.021 (0.076)	0.003 (0.068)	0.033 (0.043)	0.095 (0.103)	−0.170 (0.129)	−0.402 *** (0.149)
RiskP-F								0.153 * (0.091)	0.025 (0.101)
RiskP-H								−0.048 (0.117)	0.239 * (0.141)
AdaptP-E								0.091 (0.120)	−0.160 (0.138)
AdaptP-S								0.351 * (0.190)	0.329 (0.204)
AdaptP-C								0.028 (0.075)	0.151 * (0.087)
Pseudo R^2^	0.091	0.116	0.019	0.034	0.055	0.073	0.040	0.134	0.165

Note: (1) Standard errors are in parentheses; (2) ***, **, * denote statistical significance at the 1%, 5%, and 10%, respectively.

**Table 5 ijerph-19-06718-t005:** The calculation of mediating effects and Bootstrap test.

Mediation Path	Indirect Effect	Direct Effect	The Proportion of Mediation Effect	Bootstrap Confidence Interval			
				Lower 2.5%	Lower 5%	Upper 5%	Upper 2.5%
Factor 1–AdaptP-S–Behav-M	0.031 **	0.121	20.38%	0.004	0.008	0.071	0.079
Factor 1–RiskP-H–Behav-A	0.019 *	0.153	11.04%	0.000	0.003	0.055	0.063
Factor 3–AdaptP-C–Behav-A	0.023 *	0.178	11.42%	−0.002	0.002	0.062	0.072

Note: **, * denote statistical significance at the 5%, and 10%, respectively.

**Table 6 ijerph-19-06718-t006:** The robustness check of the regression.

Variables	Behav-M		Behav-A	
	Model 10	Model 11	Model 12	Model 13
Factor 1	0.066 **(0.029)	0.054 *(0.150)	0.066 **(0.026)	0.049 *(0.026)
Factor 2	0.100 ***(0.029)	0.105 ***(0.030)	−0.018(0.026)	−0.013(0.026)
Factor 3	0.040(0.029)	0.019(0.029)	0.082 ***(0.026)	0.066 **(0.027)
Factor 4	−0.032(0.029)	−0.036(0.030)	−0.066 **(0.026)	−0.071 ***(0.026)
Risk-F		0.034 *(0.029)		0.004(0.018)
Risk-H		−0.010(0.020)		0.039 *(0.023)
AdaptP-E		0.019(0.026)		−0.028(0.024)
AdaptP-S		0.078 *(0.027)		0.061(0.038)
AdaptP-C		−0.006(0.043)		−0.027 *(0.015)
Constant	0.442 ***(0.029)	0.219(0.017)	0.273 ***(0.026)	0.234 *(0134)

***, **, * denote statistical significance at the 1%, 5%, and 10%, respectively.

## Data Availability

Data is contained within the article or Appendix A.

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
