# Peer review of "Farmers’ Adaptive Behaviors to Heavy Metal-Polluted Cultivated Land in Mining Areas: The Influence of Farmers’ Characteristics and the Mediating Role of Perceptions"

_ijerph, 2022, doi:10.3390/ijerph19116718_

Round 1
Reviewer 1 Report
Thank you for the opportunity to read an interesting article on the importance of the perception of the impact of heavy metals in agriculture from the social point of view, i.e. by farmers working in specific conditions of agricultural production in the area of mining excavations and heavy metal pollution. The problem is described by the authors as significant in this particular region, but for a reader from another part of the world there is no information about the scale of the phenomenon, both in China and comparisons with other countries that carry out intensive extraction of minerals (based on research by other authors) and conduct agricultural production . In the opinion of the reviewer, this should be completed in the study.
The study is logical, but with regard to the description, it seems necessary to supplement the interpretation, e.g. it concerns table 2.
In addition, I propose to indicate how the surveyed sample relates to all farms in the villages surveyed (to emphasize the representativeness of opinions) and to transfer the tables from the appendix to the main text as the characteristics of the respondents and the presentation of agriculture in areas contaminated with heavy metals
Detailed comments
There are repetitions and gaps in the text v. 126 and 127, repetition of verse 124 and 125
v. 503 "insert reason here ???" - there was no indication of a reason
Generally, with regard to tables, units of measurement are usually missing and their titles require more detail and supplementation.
Reviewer 2 Report
This study investigated the adaptive behaviour of a sample of Chinese farmers to heavy metal pollution, highlighting the mediating role played by risk perception. The study suggests that proper risk training is important for promoting adaptive behaviour.
The selection of participants was careful. The analysis method could be applied to other similar situations, to verify whether the perception of risk plays the same role in other cases as well.
The article is missing the "Limitations" session. The authors did not indicate whether participation in the survey was mandatory, as appears to be considering the very high number of responses. Authors should also indicate whether farmers were assisted in responding. In addition, it would be useful to know if the evaluation of an ethics committee has been requested.
